# Age-Related Changes in Predictors of BMI in 6, 9 and 12-Year-Old Boys and Girls: The NW-CHILD Study

**DOI:** 10.3390/jfmk10030320

**Published:** 2025-08-18

**Authors:** Barry Gerber, Anita Elizabeth Pienaar

**Affiliations:** Physical Activity, Sport and Recreation (PhASRec), Faculty of Health Sciences, Potchefstroom Campus, North-West University, Potchefstroom 2520, South Africa; anita.pienaar@nwu.ac.za

**Keywords:** BMI, covariates, predictors, gender differences, longitudinal

## Abstract

**Background**: Information on childhood body composition is critical to understanding children’s growth, development, and long-term health outcomes. BMI metrics, however, have several limitations for assessing and understanding changes in BMI. Therefore, understanding the influence of various body composition factors (covariates) that are linked to, and influence, BMI over time in growing children is important. This study aims to determine sex differences in longitudinal changes in covariates of BMI from 6 to 13 years. **Methods**: Participants (N = 332, 160 boys 172 girls) from North West Province in South Africa were assessed longitudinally at the following three time-points during their primary years of schooling: Grade 1 (6–7 years); Grade 4 (9–10 years); and Grade 7 (12–13 years). Covariates included: stature (cm); body weight (kg); sub-scapular-, calf-, and triceps skinfolds (mm); body fat percentage (%), relaxed forearm, waist and mid-upper arm circumferences; percentage fat weight; and percentage muscle weight. Correlational analysis and multiple stepwise regression analysis in SPSS analyzed the significance of the contributions of the different covariates to changes in BMI from 6 to 12 years. **Results**: Different covariates influence BMI in boys and girls at different ages and the covariates also change over time in boys and girls. Weight had the strongest influence on the BMI of boys and girls, although the prediction value decreased over time. Weight and stature were consistently the strongest BMI predictors across all ages in boys. In girls, a broader range of variables influences BMI from a younger age, where slightly higher BMI correlations with fat-related variables emerged, and the percentage of fat weight distribution was a strong influential factor. These findings indicate a more in-depth analysis of BMI to determine sound intervention strategies.

## 1. Introduction

Obesity in children and adolescents has become a major global issue over the past decade [1], resulting in rising global health issues, such as noncommunicable diseases, especially in low- to middle-income countries (LMICs) [2]. Recently, the World Health Organization [1] reported that 37 million children under the age of 5 were overweight in 2022, with 390 million children and adolescents aged 5–19 years being overweight in 2022, including 160 million who were living with obesity. Various studies have globally indicated that being overweight and obese is associated with detrimental health effects, with carry-over effects from childhood to adulthood [3,4,5]. Therefore, providing effective and compassionate care tailored to the child and family, focusing on healthy body composition, is vital [6].

Body composition is defined as the proportional composition of a person’s total body weight, which consists of muscle, bone, fat, and other tissue [7]. Fat weight refers to the extractable fat in adipose and other body tissues, while fat-free body weight can be defined as the remaining fat-free chemical substances and tissues in the body, which include muscle, bone, connective tissue, and internal organs [8]. A pressing issue nowadays is determining the best measuring tool or protocol with which to assess body composition in children and adults. In this regard, various high-technology methods for measuring body composition are available, including dual-energy X-ray absorptiometry, air displacement plethysmography, and hydrostatic weighing. However, these methods are not suited for field-based testing and are expensive. Although other methods, such as bioelectrical impedance analysis (a technique grounded in the basic principles of electrical conductivity, introducing a small, weak alternating electrical current into the body at one or more frequencies via electrodes) and the widely used body weight index (BMI) over decades, are more practical, inexpensive, easy to use and affordable, the accuracy of these methods is still in question [9,10,11,12]. In this regard, various studies globally attest to the reliability, precision, and accuracy of impedance technologies and have found correlations greater than 0.95 with dual energy X-ray absorptiometry (DXA) or dilution-measured TBW with regards to TBW or FFM, although the level of agreement may be large (±5–10%) [9,13,14]. Furthermore, Feng et al. [15] found high correlations between BIA and dual-energy X-ray absorptiometry (DXA) measurements for fat weight (FM) and fat-free weight (FFM). However, BIA tended to underestimate FM by approximately 1.84 kg and overestimate FFM by about 2.56 kg. Another study, episodically focusing on children with obesity, revealed that multi-frequency octopolar BIA devices closely matched DXA results, with minimal differences in body fat percentage and lean weight; however, single-frequency devices significantly underestimated body fat percentage [16]

BMI, which is an indirect method for assessing body composition using only stature and weight variables, fails to distinguish between fat, muscle, and bone mass; consequently, it is prone to misclassification, particularly among individuals with a muscular build [17]. Various other researchers have also indicated over the past decade that the BMI metric has several limitations for assessing reasons for changes in BMI [18,19,20,21]. In this regard, the relationship between BMI and BMI z-scores (BMIz) is curvilinear, approaching a maximum value that varies by sex and age [22]. Therefore, Freedman et al. [22] believe that it is crucial to understand how BMI fluctuates over time in growing children to determine the most suitable scale for measuring BMI changes. Furthermore, it is essential to understand the covariates of BMI that influence these fluctuations and changes in BMI over time [23].

Sattar et al. [24] indicated that although BMI is based on two variables (stature and weight), it can be influenced by various covariates, including body fat, muscle weight, sex, race, age, and waist circumference. In this regard, referring to changes in specific covariates of BMI, the WHO guidelines indicate that at 5 years of age, the differences between the 15th and 85th percentiles for sub-scapular and triceps skinfolds are 3 mm and 4–5 mm, respectively [25]. Furthermore, Wernech et al. [26] found that the sum of skinfolds (triceps and sub-scapular) of boys and girls between 7 and 10 years old changes from 23.5 mm to 33.5 mm (boys) and 25.7 mm to 33.9 mm (girls). Regarding fat percentage, Cossio-Bolaños et al. [27] report that although they found small differences in fat percentage of 2.1% (girls) and 9% between the ages of 7.5 and 15.4 years, significant standard deviations (SD) were found within each year group, of up to 6.9% (girls) and 8.7%(boys). Lalucci et al. [28] investigated the correlations between and covariates and found correlations between BMI and fat to lean weight ratio (general, r = 0.69; female, r = 0.74; male, r = 0.69); BMI and fat-free weight (general, r = 0.49; female, r = 0.67; male, r = 0.44); BMI and skeletal muscle weight (general, r = 0.50; female, r = 0.68; male, r = 0.44); and BMI and body fat percentage (general, r = 0.47; female, r = 0.54; male, r = 0.40; all with *p* < 0.01). These results underline the theory that there is more to just BMI. As a consequence, in order for researchers to understand changes in BMI better, further investigation is needed to verify if the covariates of weight (body fat percentage, skinfolds, circumferences) play a vital role in BMI changes, or if the ratio of the percentage of change from the covariates changes over time, especially during childhood, as children grow and mature. Furthermore, it is essential to investigate if the covariates influence BMI differently in boys and girls. In this regard, Mast and coworkers [29] had already found in 1998 that WHR, the sum of four skinfolds and fat percentage, differed German children between 5 and 7 years of age. However, we found no studies that specifically focus on possible changes and influences of various anthropometric and body composition covariates concerning BMI, and whether this difference might, in turn, affect the BMI of different genders differently. Considering the above studies in the literature, and based on various methods that are available to determine body composition, including BMI, the accuracy of the BIA method in determining BMI and fat and muscle percentages in children was first analyzed as a sub-objective to decide whether we could use the additional body composition measures available, including muscle mass and fat mass, in our analysis. As a primary objective, this study aims to determine whether changes in the covariates of BMI occur in 6-, 9-, and 12-year-old boys and girls, and whether such changes differ between boys and girls.

## 2. Materials and Methods

### 2.1. Research Design

This research is a sub-study of the NW-CHILD (Child-Health-Integrated-Learning and Development) longitudinal study conducted in Northwest Province (NWP) in South Africa between 2010 and 2016. This longitudinal study had three time-points (2010, 2013, and 2016) during the primary school years. The primary research included demographic questionnaires, physical fitness, physical activity, anthropometric, physiological, and blood pressure measurements.

The NW-CHILD study employed a random and stratified sampling method to select districts, schools, and participants, stratified by gender and school quintile. As part of the design, schools were randomly selected from four of the eight districts within NWP. Five schools from each district were selected, totaling 20 schools, for the testing. Participants from NWP were assessed at three time-points during their primary years of schooling: Grade 1 (6–7 years); Grade 4 (9–10 years); and Grade 7 (12–13 years). Schools from each district represented a range of socioeconomic statuses, with schools from quintiles 1–3 categorized as low SES and those from quintiles 4–5 considered high SES, according to South Africa’s classification system [30].

The NW-CHILD study received ethical clearance from North-West University’s Health Research Ethics Committee (HREC) (00070-09-A1) and approval from the North-West Department of Basic Education (DBE). The participating school principals gave consent for testing during school hours and parents of all students were provided with an informed consent form. Additionally, children whose parents gave consent to their participation were required to give assent if they were under 8 years old or consent if they were older.

### 2.2. Investigating Group and Procedures

The initially recruited 860 participants, who were enrolled from 20 primary schools in 2010 (Grade 1) at baseline, and were approximately six years (±0.39) of age. Although 829 parents consented for the children to participate, on the day of testing, only 816 (419 boys and 397 girls) were available due to absence on the day of testing or exclusion because of inaccurate ages. For the first follow-up measurement, 574 participants (282 boys and 292 girls; dropout rate 29.7%) aged approximately 9 years (±0.38) consented to participate in the study. At the final time-point measurement in 2016, 381 participants (181 boys and 200 girls) consented, indicating a further drop-out of 33.6%. Consequently, the study had a total dropout rate of 57.2% (n = 467) over the 6-year follow-up period. Furthermore, although some participants were present at all time-points, others had incomplete data due to various reasons. Consequently, another 49 participants were excluded, resulting in a final group of 332 (38.6% of the initial participants) participants (160 boys and 172 girls). This study focused on all participants who participated in all three time-point measurements in 2010, 2013, and 2016. A previous study by Pienaar (2015) [31] from the same research group investigated possible bias due to lost subjects during follow-up analysis using *t*-tests. Insignificant Cohen’s d values provided no evidence of bias in baseline height (*p* = 0.553, d = 0.04), mass (*p* = 0.03, d = 0.16), BMI (*p* = 0.008, d = 0.19) and fat percentage (*p* = 0.223, d = 0.09). All children who presented with physical disabilities and whose parents did not consent were excluded. The layout and set-up of the stations were kept as similar as possible but varied between school settings according to the available space.



### 2.3. Measurement Instruments and Apparatus

Anthropometric measurements

Anthropometry, which included stature (cm), body weight (kg), measurement of the sub-scapular, calf, and triceps skinfolds (mm), relaxed forearm and waist circumferences (cm), was measured before any fitness testing commenced. The fat percentage was calculated by adding the triceps and sub-scapular skinfolds and assessed according to the gender-specific equation developed by Lohman (1992) [32].

All anthropometric tests were conducted by qualified level 2 Kinantropometrists who were postgraduate students specializing in Kinderkinetics following International Society for the Advancement of Kinanthropometry (ISAK) protocols [33].

A Harpenden portable stadiometer (Holstein Limited, London, UK) and two electronic scientific calibrated scales, including the SECA and Omron BF 511 [34], measured stature and body weight to the nearest 0.1 cm and 0.1 kg. BMI was calculated from these measurements (body weight in kg divided by length in m^2^).

Body composition

The Omron BF 511 BIA body composition analyzer (OMRON, Milton Keynes, UK) was used to measure body composition characteristics, including fat-free weight (percentage muscle weight) and body fat (percentage body fat). The rationale for using the OMRON BF511 is that it is a non-invasive, quick, and easy-to-use clinically validated bioelectrical impedance analysis (BIA) device that offers a portable and cost-effective alternative to expensive methods, and can be operated with minimal training, making it ideal for use in community and school environments. Its application in pediatric populations, particularly in children aged 6 years and older, is supported by several methodological and practical advantages. No funding was received from the industry for using the equipment.

In this regard, participants wore minimal clothing, no jewelry that could influence the bioimpedance, and were barefoot. The participants stood on the Omron scale, positioned on the indicated footpads and held the hand devices with outstretched arms in front of their bodies while the measurements were taken. The mid-upper arm and waist circumferences (in cm) were measured using a metal measuring tape (Cescorf, Triteza, Brazil) and taken twice to ensure validity and reliability. The sub-scapular, calf, and triceps skinfolds (mm) were also taken twice with a Harpenden skinfold caliper to provide an average value. This takes into consideration technical errors in measurement.

If necessary, translators were made available when English or Afrikaans was not the participant’s first language.

### 2.4. Statistical Analysis

The “Statistica for Windows” [35] and Statistical Package for the Social Sciences (SPSS) for Windows, version 27 [36], were used. A previous study by Pienaar [31] analyzed the same sample population and indicated that no bias was introduced by the loss of subjects over time. The data were assumed to be missing at random and did not significantly affect the results. Diagnostic checks were performed to ensure that the assumptions of linear regression were met, including the evaluation of residual plots for normality, linearity, and homoscedasticity. Standardized residuals were examined, and plots of residuals versus predicted values did not indicate patterns suggestive of heteroscedasticity.

Firstly, a collinearity diagnostics analysis was performed on the baseline measurements (T1) (see Appendix A), with results revealing a condition index above 30 in the final dimension, suggesting potential multicollinearity. The variance proportion analysis identified that mid-upper arm circumference is strongly associated with the final dimension (variance = 0.98), indicating it may be collinear with the model intercept. A moderate multicollinearity between triceps and calf skinfolds was also found. To address this, a principal components analysis (PCA) of the body composition variables (excluding weight and stature) was performed to assess dimensionality and address multicollinearity (see Appendix A). Results indicated a significant Kaiser-Meyer-Olkin (KMO) value of 0.818, and Bartlett’s Test of Sphericity (χ^2^ = 1412.94, *p* < 0.001), indicating that the data were suitable for PCA. However, the 2 new loadings including component 1, adiposity (dominated by skinfolds and fat %), and component 2, muscle mass (loading almost exclusively on muscle %), were deemed not to be more interpretable predictors than the original variables, and it was therefore decided to continue with the original variables to investigate how these different body composition variables, although closely linked to each other, might contribute differently to changes in BMI over a longitudinal period.

Descriptive statistics were calculated to analyze means and standard deviations of the boys’ and girls’ age and anthropometric and body composition profiles. A correlation coefficient analysis was conducted to determine the possible correlations between BMI and body Weight and anthropometric and body composition covariates, including stature, Weight, sub-scapular skinfold, triceps skinfold, calf skinfold, waist-circumference, mid-upper arm circumference, percentage fat mass, and percentage muscle mass. Cut points used to indicate significance included 0.1 < r < 0.3 (small correlation), 0.3 < r < 0.5 (medium correlation), and r > 0.5 (strong correlation) [37]. A multiple stepwise regression analysis was conducted in SPSS to determine the significance of the contributions of the different covariates to changes in BMI. The following cut points indicate the percentage variance explained: R2 = 1% is interpreted as a small effect, R2 = 10% as a medium effect, while R2 ≥ 25% is considered a large effect. For statistical significance, *p* is set at ≤ 0.05.

## 3. Results

The study included 332 participants between the ages of six and thirteen, including 160 boys and 172 girls, with mean ages of 6.79 + 0.50 (Grade 1, T1), 9.89 + 0.38 (Grade 4, T2), and 12.90 + 0.38 (Grade 7, T3), at each of the three time-points, who had complete datasets. Measurements taken over the 7-year longitudinal period during the primary school years provided 6-year follow-up data for this study.

Before analyzing the contributions of different body composition characteristics to BMI at various ages by using regression analysis, a correlational analysis was performed to determine the association between direct (BMI equation and sum of skinfolds) and indirect estimates of BMI derived from bioelectrical impedance analysis, conducted with the Omron BF 511 body composition analyzer.

In this regard, the results from Table 1 indicate a high correlation at T1 (r > 0.93) between the direct and indirect methods to determine BMI. This association strengthened over time (r = 0.97, T2; r = 0.99, T3). Regarding the correlation between the fat percentage determined by skinfolds and BIA, a slightly lower, yet still strong, association was found between the two methods (T1: r = 0.68; T2: r = 0.89; T3: r = 0.86). As a high association was evident from both analyses, taking into consideration that both methods have limitations, BIA results obtained from the Omron BF511 were considered to demonstrate consistency in output between the two methods. Hence, for this study, BMI, fat percentage, and muscle percentage, as measured by the Omron BF511, were used in all further analyses.

Table 2 reports the changes and significance of changes (*p* < 0.01) in age and body composition measurements across the three time-points (T1: 6 years, T2: 9 years, T3: 12 years) for boys (N = 160) and girls (N = 172). Similar age distributions were observed at each time-point, with boys being slightly older at each time-point; however, this difference was insignificant (*p* > 0.05).

The boys’ anthropometric and body composition profiles over the three time-points (Table 2) revealed statistically significant changes (*p* < 0.01) across most variables. Body mass index (BMI) increased from 15.95 ± 2.25 kg/m^2^ at T1 to 19.61 ± 4.34 kg/m^2^ at T3, reflecting a significant cumulative gain of 3.66 kg/m^2^. Similarly, stature and body weight showed continuous and statistically significant increases over time, with stature increasing by 31.75 cm and body weight increasing by 22.6 kg from T1 to T3. Sub-scapular and calf skinfolds showed consistent and significant increases at all time-points, while triceps skinfold thickness increased significantly from T1 to T2 (+2.08 mm) and from T1 to T3 (+2.77 mm), but not between T2 and T3 (+0.69 mm, *p* > 0.01). Waist circumference increased from 55.13 ± 6.01 cm to 66.40 ± 10.03 cm across the three measurements, with significant changes (*p* < 0.01) observed at each interval. Although mid-upper arm circumference was only recorded from T2 onwards, it increased significantly (++2.36 cm), from 20.65 cm to 23.01 cm by T3 (*p* < 0.01).

Regarding body composition, boys demonstrated a significant increase in muscle mass percentage, rising from 25.23% at T1 to 35.83% at T3, indicating a total gain of 10.6% over the six years. However, examining the fat mass percentage revealed minor changes, with only the increase from T to T2 (+1.86%) being significant (*p* < 0.01). Furthermore, a non-significant decrease from T2 to T3 was found in fat mass percentage, resulting in a slight, insignificant overall increase of 0.8%.

Significant changes were also observed in girls across all anthropometric and body composition measures from 6 to 12 years of age. BMI changed significantly, with 4.59 kg/m^2^ from 15.69 ± 2.02 kg/m^2^ at T1 to 20.28 ± 4.11 kg/m^2^ at T3 *p* < 0.01). Stature and weight showed even greater increases than those observed in boys, with stature rising by 35.02 cm and weight increasing by 26.09 kg over the study period. These changes were statistically significant across all time intervals. Subcutaneous fat levels measured by skinfolds increased significantly across all measurement sites for girls. Triceps, sub-scapular, and calf skinfolds all showed significant gains (*p* < 0.01) at each interval, with the calf skinfold demonstrating the highest overall increase (+9.08 mm). Waist circumference increased from 54.03 ± 4.90 cm at T1 to 66.21 ± 8.85 cm at T3, with statistically significant changes across all time-points.

Mid-upper arm circumference, measured from T2 onward, increased significantly (*p* < 0.01), by 3.07 cm, at T3. Girls also exhibited significant increases in body fat percentage, rising from 16.41% at T1 to 25.45% at T3, with each interval showing statistically significant changes (*p* < 0.01). Muscle mass percentage increased significantly, albeit to a lesser extent than in boys, rising from 26.62% to 32.71% over the six years (+6.09%, *p* < 0.01).

The increasing standard deviations observed across most variables from T1 to T3 in both boys and girls indicate growing variability in body composition with age. For both sexes, measurements, such as weight, skinfold thickness, and body fat percentage, showed wider distributions over time, reflecting individual differences in growth rates, maturation timing, and lifestyle factors. Notably, girls exhibited slightly higher variability in fat-related measures, while boys showed greater dispersion in muscle mass gains. These findings highlight the importance of considering inter-individual variability when interpreting developmental trends during late childhood and early adolescence.

Table 3 and Figure 1 and Figure 2 display the results of a correlation analysis between the various body composition variables, weight, and BMI at the three age time-points by sex. The correlation patterns between body composition variables and weight and BMI show notable sex differences across the three measurement points. In boys, stature maintains a stronger relationship with weight than BMI, while in girls, this association is weaker overall, especially with BMI. Skinfold measures (sub-scapular, triceps, and calf) and waist circumference show consistently strong correlations with both weight and BMI for both genders. However, girls generally exhibit slightly higher BMI correlations with fat-related variables, notably with the percentage of fat mass. Muscle mass presents a contrasting trend; while it initially shows a moderate positive correlation with weight in boys and girls, this shifts to a negative correlation with BMI over time, which is more pronounced in girls.

Overall, while BMI and weight are both strongly correlated with several body composition variables, BMI appears to be a more sensitive indicator of fat accumulation and a less reliable measure of muscularity, especially in girls as they age.

BMI included the sub-scapular skinfolds, triceps skinfolds, calf skinfolds, and waistcircumference, which all showed strong positive correlations across the three time-points, with correlations often exceeding 0.75. Mid-upper-arm circumference also displayed strong associations, particularly at T2 and T3, which correlated highly with weight and BMI.

Body fat percentage demonstrated moderate to strong correlations with weight and BMI, with stronger associations observed at later (older) time-points (9 and 12 years). Conversely, the percentage of muscle mass exhibited weak or negative correlations, particularly at 12 years, suggesting that it has a minimal influence on either weight or BMI. Overall, body fat measurements (skinfolds and fat percentage) consistently showed stronger relationships with BMI than with muscle mass, emphasizing the greater role of fat in predicting both weight and BMI across time.

In girls, the correlations reveal several key patterns. Weight and BMI exhibit a strong positive correlation at all time-points, with the strongest relationship observed at 12 years (r = 0.93). Body fat measurements, including sub-scapular, triceps, and calf skinfolds, consistently show moderate to strong correlations with both weight and BMI, particularly at later time-points (9 and 12 years), highlighting their significant role in predicting body composition. Waist circumference demonstrates the strongest association, with correlations exceeding 0.90 for both weight and BMI at 9 and 12 years. The percentage of fat mass also shows a strong positive correlation with BMI (ranging from 0.88 to 0.95) across all time-points, while also correlating moderately with weight. In contrast, muscle mass percentage exhibits weak or negative associations with BMI and weight correlations, particularly with both weight and BMI, especially at 9 and 12 years, indicating a diminishing relationship over time. Overall, fat-related variables showed stronger correlations with both weight and BMI, while muscle mass showed a limited influence, particularly with increasing age.

Figure 1 and Figure 2 illustrate changes in correlations, revealing that BMI maintained consistently strong associations with subcutaneous fat measures (e.g., skinfolds, waist circumference) and fat mass percentage across all ages, indicating its continued relevance as a proxy for adiposity during childhood and early adolescence. However, the strength of association between BMI and muscle mass declined over time, becoming negative by age 12, particularly in girls, highlighting a shift in BMI’s predictive utility. These findings underscore the importance of considering developmental stage and sex when interpreting BMI in relation to underlying body composition.

Table 4 and Table 5 provide the results of a series of stepwise regression analyses conducted to examine the contribution of different body composition traits to BMI across different ages. Each regression model included the following independent variables: stature; weight; sub-scapular, triceps, and calf skinfolds; waist circumference; mid-upper arm circumference; fat mass percentage; and muscle mass percentage. In the multiple regression analysis, β (beta) coefficients were used to assess the independent contribution of each body composition variable to BMI while controlling for the effects of other predictors. Positive β values indicated that increases in a given variable were associated with a higher BMI. In contrast, negative β values reflected an inverse relationship, such as those observed in values for stature. The magnitude of the β coefficient denotes the strength of the association, with larger absolute values indicating stronger predictive power. These coefficients offer a clearer understanding of which variables most significantly influence BMI across different developmental stages.

The multiple regression analysis for boys (Table 4) across the three age time-points reveals shifting patterns in the predictors of BMI as boys progress through early to mid-childhood. Strong relationships between the dependent and independent variables were found across all three age models (6, 9, and 12 years). In this regard, a correlation coefficient between R = 0.9458 and R = 0.9876 was found, with a coefficient of determination between R^2^ = 0.8946 and R^2^ = 0.9755. This indicates a good-to-excellent fit, explaining between 89.46% and 97.55% of the variance in these dependent variables, with the highest fit at age 9. All three models are statistically significant (*p* < 0.0000), supported by high F-statistics between 160.32 and 644.95, reinforcing the robustness of the models. At age 6 (T1), only two variables—weight (b* = 1.398) and stature (b* = −0.773)—showed significant contributions (*p* < 0.001). By age 9 (T2), the model’s explanatory power increases substantially (R^2^ = 0.976) and more variables emerge as significant predictors. In addition to weight and stature, waist circumference, fat percentage, mid-upper arm circumference, and calf skinfold also become significant. At age 12, although the overall variance explained slightly decreases (R^2^ = 0.964), the pattern becomes more refined, with waist circumference (b* = 0.324) becoming the strongest contributor after weight. At the same time, fat percentage and mid-upper arm circumference also remain significant, while the sub-scapular skinfold becomes a modest but significant predictor (b* = 0.067, *p* = 0.028). Notably, muscle mass percentage and most skinfold measures remain statistically nonsignificant across all ages.

In girls (Table 5), the regression analyses reveal strong relationships between the independent variables and the dependent variable across the three age models. For Model T1 (age 6), the correlation coefficient (R = 0.9962) and coefficient of determination (R^2^ = 0.9923) indicate an excellent fit, explaining 99.23% of the variance in the dependent variable. Similarly, Model T2 at age 9 (R = 0.9591, R^2^ = 0.9199), and Model T3 at age 12 (R = 0.9881, R^2^ = 0.9764), also demonstrate strong explanatory power, with R^2^ values above 0.91. All three models are statistically significant (*p* < 0.0000), supported by high F-statistics (2634.5, 206.81, and 744.33, respectively), reinforcing the robustness of the models. The standard errors of the estimates are relatively low, with the smallest error (0.18146) at age 6, followed by ages 12 (0.64973) and 9 (0.99390), suggesting that the models’ predictions are highly accurate. Notably, several independent variables consistently demonstrate statistically significant relationships across the models. Stature (b* = −0.585361, *p* < 0.0001 for T1) and fat percentage (b* = 0.053296, *p* = 0.000385 for T1) have strong and significant effects on the dependent variable, with negative and positive relationships, respectively.

The effect of weight is consistently significant and positive across all models (b* ranging from 0.412900 to 1.061365). However, other variables, such as the sub-scapular and triceps skinfolds, show less consistent significance.

At age 9, the triceps skinfold is significant (b* = −0.173048, *p* = 0.005272); however, at age 12, it becomes non-significant (b* = 0.002438, *p* = 0.939485). These findings suggest that while some variables, such as body fat percentage and stature, consistently influence the outcome, others may have context-dependent effects that vary across models. Overall, the results demonstrate the predictive power of the models, highlighting the importance of specific variables, such as stature, body fat percentage, and weight, in explaining the variations in the dependent variable.

When exploring sex differences in the predictive power of the independent variables and changes in the predictive power, as displayed in Figure 3 (boys) and Figure 4 (girls), the models demonstrate strong predictive power, with R^2^ values consistently above 0.89 for boys and 0.91 for girls, indicating that the included variables explain a significant proportion of BMI variance. Weight (weight) emerges as the strongest predictor in both boys and girls, although its influence is initially higher in boys (β* = 1.40 at age 6) and stabilizes over time. In contrast, the impact of stature (height) is consistently negative, with a stronger effect on boys (β* = −0.77 at age 6) compared to girls (β* = −0.58 at T1).

Fat percentage significantly predicts BMI in both groups, but with a greater effect in girls, especially at age 9 (β* = 0.36, *p* < 0.001). Mid-upper arm and waist circumference measurements become increasingly relevant for boys over time, whereas they show earlier significance in girls. Overall, BMI prediction is slightly stronger in girls, particularly at age 6, suggesting sex-based differences in body composition dynamics. These findings highlight the importance of considering both adiposity and muscle distribution when modelling BMI trajectories in children.

## 4. Discussion

The primary objective of this study was to investigate potential changes in the associations between body composition covariates, specifically BMI, from 6 to 12 years in boys and girls. Taking into account the evidence of potential multicollinearity found as explained, the main finding indicated that weight and stature were consistently the strongest predictors of BMI in both sexes, particularly at younger ages. As children aged, additional variables, such as waist circumference, mid-upper arm circumference, and fat percentage, became more influential, with differences in the timing and magnitude of these effects between boys and girls. Girls showed a broader set of significant predictors earlier in development, while boys exhibited a more consistent pattern over time. These findings highlight sex-specific developmental trajectories in body composition and BMI determinants. In this regard, no studies were found to compare these results with, and no study specifically focuses on the changes and influences of various anthropometric and body composition covariates regarding BMI and sex differences. As a sub-objective, we first analyzed the validity of the OMRON BF 511 bioimpedance scale to determine BMI and fat percentage, as we were interested in body composition variables that are only available with this measuring tool. High correlations (r > 0.93) were found between the OMRON-generated BMI and BMI-calculated scores (Table 1). The validity of the Omron to determine other aspects of body composition, and more specifically the percentage of BF, from the Omron BF 511, further indicated significantly high correlations (0.68 > r < 0.89) with BF, as calculated with an equation. In agreement, another study conducted by Brtková [38], which focused on 52 older participants with an average age of 22.4 years, also found an even stronger correlation (r = 0.93) between skinfold measurements and the use of the Omron BF511 as parameters for body fat percentage. These results align with the parameters set out by Omron stipulating that the accuracy of Omron models as bioimpedance scales varies from model to model based on the ‘Standard Error of Estimate’ (SSE). The SSE stipulates that 68% of all measurements for different users are accurate to within 3.5–4.1%, relative to body fat percentage (kg.m^2^) [34]. Our results agreed with this and verified that the OMROM B511 bioimpedance analyzer could be used in our study as an accurate measuring tool with which to determine the BMI and body composition of children relatively accurately.

Although the aim of this paper was not to investigate sex differences in anthropometric and body composition profiles (Table 2), boys and girls showed similar anthropometric profiles throughout the study, with boys being slightly taller at age 6 (+2.6 cm) and T2 (+0.01 cm), with girls surpassing them at age 12 (+0.67 cm). Similar results were found in weight, although girls surpassed boys at age 9. Consequently, these minor differences resulted in very similar BMI scores (0.04–0.06) in boys and girls throughout the study. These results align with an Australian study by Cochrane [39], which also found only slight differences between boys’ and girls’ anthropometric profiles between the ages of 6 and 13. It is, however, not expected that significant anthropometric differences will exist before the onset of puberty due to insignificant hormonal influences [40]. Regarding body composition, girls tend to show higher body composition scores regarding skinfolds and body fat percentage throughout the study, from age 6 to 12 years (Table 2). In this regard, Pelemis et al. [41] found that girls at an average age of 6.25 years already showed higher abdominal and skinfolds at the back than boys. Furthermore, earlier research already showed that girls tend to have higher body fat percentages and lower muscle mass percentages compared to boys before puberty [29], which in turn will be noticeable in their slightly higher circumferences and skinfolds.

The body composition characteristics of young developing children are critical to understanding growth, development, and long-term health outcomes. Differences in body fat percentage, lean weight, and fat distribution between boys and girls emerge during primary school and become more pronounced during puberty, according to various studies [42,43,44], which our findings also confirmed. Although both boys and girls exhibit steady increases in height and weight throughout childhood, body composition diverges as they approach puberty [45]. Boys develop greater lean mass and bone density, whereas girls tend to accumulate more fat mass, particularly in the gluteofemoral region [46]. These differences are primarily driven by hormonal changes, particularly by increases in *estrogen* in girls and *testosterone* in boys [43]. Results from the current study are well aligned with this; boys initially had a higher fat percentage and lower muscle mass percentage compared to girls at age 6; however, this changed at a later age (Table 2). At an average age of 9 years, boys already surpassed girls in terms of muscle mass percentage (*testosterone*-related), with the difference becoming even larger with increasing age. On the other hand, boys also initially had a lower fat percentage at age 6, with girls surpassing them from age 9, with sex differences increasing further at age 12. This might be due to girls entering the pubertal phase with increased *estrogenic* influences.

Our sex-specific analysis, which identified the body composition variables that influence BMI at 6, 9, and 12 years and whether these contributions varied by age, revealed notable differences between boys and girls (Table 3, Table 4 and Table 5 and Figure 1, Figure 2, Figure 3 and Figure 4). For boys, weight and stature were consistently the strongest predictors across all time-points, with waist circumference and fat mass percentage gaining predictive strength with age. By age 12, the mid-upper arm circumference also became a significant indicator. Skinfolds and muscle mass percentage generally showed weak or non-significant contributions. In contrast, a broader range of significant predictors emerges in girls, particularly at younger ages. At the age of 6, fat mass percentage, muscle mass percentage, and sub-scapular skinfold thickness significantly influence BMI, alongside weight and stature. By the ages of 9 and 12 years, like boys, waist circumference, fat mass percentage, and mid-upper arm circumference become more influential, while skinfolds and muscle mass percentage lose significance. Overall, the BMI of girls is more strongly influenced by fat distribution and composition at younger ages, whereas the BMI of boys is more consistently driven by weight and stature. These patterns suggest developmental and physiological differences between boys and girls in terms of fat accumulation and body composition. The findings align with, and add to, existing studies in the literature on body composition, particularly regarding fat mass, lean muscle mass, and growth patterns [47,48].

The models consistently indicate that weight is the strongest and most significant predictor of body composition in both sexes throughout the study. However, the strength of this contribution declined by more than 60% in boys (Table 4) and by approximately 40% in girls (Table 5) between the ages of 6 and 12 years. This is consistent with studies showing that total body weight is a primary determinant of fat-free mass and fat mass in children [46,49]. Interestingly, stature (height) had a negative relationship with body composition in both genders across all three age models. This suggests that as boys and girls grow taller, they may not necessarily gain proportional fat mass or muscle mass increases in this age group. This finding is supported by research indicating that taller children tend to have lower body fat percentages relative to their weight [50,51]. However, sex differences exist, especially at a later age, as boys tend to develop more lean weight, while girls have a higher percentage of body fat [43]. Furthermore, stature was primarily found in the top three influences of BMI in boys and girls; however, in girls, stature contributed to only the 4th highest influence at age 9. This might be due to girls entering their pubertal phase at around the age of 9–10 years, with puberty influences, such as increased fat mass (as seen in Table 5), contributing more towards BMI.

In boys, waist circumference was one of the top three significant contributors in all three models (Table 4, Figure 3) compared to girls, where waist circumference was insignificant at age 6 (*p* = 0.97). Waist circumference was 6th in contributing to BMI at age 9 (*p* < 0.01) and only become a significant factor at age 12, where it was 3rd in the stepwise model (Table 5, Figure 4). This is well aligned with findings reported by Nazare et al. [52] that reveal a slightly stronger correlation between the waist circumference of boys compared to girls, with a BMI (r = 0.84) for females and (r = 0.87) for males. Neuhauser [53] reports that waist circumference in children and adolescents varies depending on age, gender, and ethnic group. In this regard, boys tend to accumulate more abdominal (central) fat than girls, especially in early childhood, where girls typically have more peripheral fat distribution (hips, thighs, buttocks), which is less directly reflected in waist circumference [54,55]. Since WC primarily measures central fat, it naturally correlates more strongly with BMI in boys. This suggests that waist circumference may reflect different types of fat in boys and girls, influencing its correlation with BMI differently across sexes. Furthermore, this also aligns with findings by Ronnecke et al. [56], showing similar trends of relatively constant increases in median percentiles from 3 to 16 years in both sexes, with boys showing higher values in all waist circumference percentile curves. Mid-upper arm circumference, although primarily located at mid-table in contribution, also emerged as a significant contributor to BMI in boys and girls from the ages of 9 to 12 years. These findings align with the findings reported by Frisancho [57] indicating that MUAC is an excellent indicator of total body fat and nutritional status in children. Since arm circumference reflects muscle and fat mass, its interrelationship with BMI is understandable, especially in girls, who tend to store subcutaneous fat in the arms [58]. The significance of MUAC, and its similar contribution compared to fat mass percentage at ages 9 and 12 years, is also confirmed by a study indicating that upper-arm measurements are more closely associated with total fat mass and lean mass distribution [51].

While fat percentage (Fat%) emerged as a significant contributor at ages 9 and 12 years in boys and girls, individual skinfold thickness measurements (Triceps sf, Calf sf, and Sub-scapular sf), although stronger in girls, show mixed results (Table 4 and Table 5, Figure 3 and Figure 4). This could be due to sex-specific and gene-related patterns of fat distribution, where girls tend to accumulate more subcutaneous fat, particularly in the gluteofemoral region. At the same time, boys have a more even distribution of fat [49]. Girls typically have higher fat mass than boys from early childhood, due to hormonal and metabolic differences, which explains why fat percentage strongly predicts BMI in girls [45].

Skinfolds are commonly used to assess adiposity. The lack of significance and/or the lower-ranked order of the individual skinfold measurements in the stepwise regression for both genders suggests that total fat percentage is a better predictor of body composition than individual skinfolds, which aligns with research by Moreno et al. [59]. The sub-scapular skinfold measurement in girls was a significant predictor at age 6 (β = 0.044, *p* = 0.0022), indicating that trunk fat plays a role in the BMI of girls. However, the triceps skinfold was negatively associated with BMI at age 9 (β = −0.173, *p* = 0.0052), which is an unusual finding, but may suggest that arm fat does not contribute significantly to BMI variations compared to trunk fat. Research by Moreno et al. [59] found, in this regard, that subcutaneous fat deposition patterns differ between boys and girls, with girls accumulating more peripheral fat and boys accumulating more central fat.

Muscle mass percentage was not a significant predictor in any model, for either boys and girls, which may indicate that muscle mass does not strongly influence the dependent variables being measured at this age. Since BMI is primarily driven by fat mass and weight during primary school years, lean mass may not significantly impact BMI calculations in boys and girls, as muscle development is not yet pronounced at this stage [45]. These findings are consistent with other studies showing that muscle mass differences between boys and girls become more significant only after puberty [49]. Boys have similar lean body mass levels to girls in early childhood but begin to develop more muscle mass as they approach puberty [46].

Because few studies were found that focused on providing an understanding of body composition co-variates that influence BMI in younger children and the changes in the contribution of these changes to BMI over a longitudinal period, our findings make a unique contribution to the scientific field to understanding changes in BMI in the age period between 6 and 13 years, and especially regarding difference in BMI changes in boys and girls in this developmental period. However, the study also has limitations that need to be considered. A significant limitation of the study that is acknowledged is the possible high degree of intercorrelation among several body composition variables, raising potential concerns regarding multicollinearity. The results from this statistical analysis would have ob-scured the ability to discern the unique contribution of each variable to the outcomes of interest. Future analyses could benefit from applying techniques such as Principal Components Analysis (PCA), to reduce redundancy and identify orthogonal compo-nents that better represent underlying constructs. Although a randomized and stratified research design was followed, the participants were recruited from only one province within South Africa, with a relatively small participant sample, which limits the generalizability of the findings to the broader South African population. Therefore, it is recommended that similar studies should be conducted in the other eight provinces for a stronger generalization. We acknowledge that not all factors that might influence BMI, such as eating habits, physical activity levels, pubertal status, and socioeconomic status, were included in our analyses. Therefore, it is recommended that future studies also investigate the influences of these covariates. Lastly, the loss of subjects over the study period was relatively large; we acknowledge that this limits the internal validity of the study findings.

## 5. Conclusions

This longitudinal study is the first to examine sex-specific changes and influences of BMI covariates in developing children aged 6 to 12 years. Weight and stature emerged as the strongest predictors of BMI across all ages, particularly at age 6. Over time, other body composition indicators—such as waist circumference, mid-upper arm circumference, and fat percentage—gained importance, with distinct patterns observed between boys and girls. Girls showed a broader range of early BMI influencers that shifted over time, while boys displayed a more consistent pattern, with increasing emphasis on fat-related variables. These findings enhance the understanding of childhood growth and body composition and have implications for early obesity screening. Furthermore, these findings highlight the complex interplay between anthropometric variables and BMI in children and the differences in this regard between boys and girls, which has implications for growth monitoring, nutritional assessment, and early interventions for childhood obesity. Furthermore, as waist circumference, mid-upper arm circumference, and fat mass percentage showed strong associations with BMI, future research should test the predictive value of these variables independently for identifying early obesity and metabolic risk, potentially replacing or supplementing BMI as a screening tool. Additionally, there is a need to translate these findings into user-friendly, cost-effective screening instruments for schools and healthcare services, particularly in low-resource settings. Future work could focus on developing and validating simplified screening algorithms that incorporate BMI covariates for the early identification of at-risk children.

## Figures and Tables

**Figure 1 jfmk-10-00320-f001:**
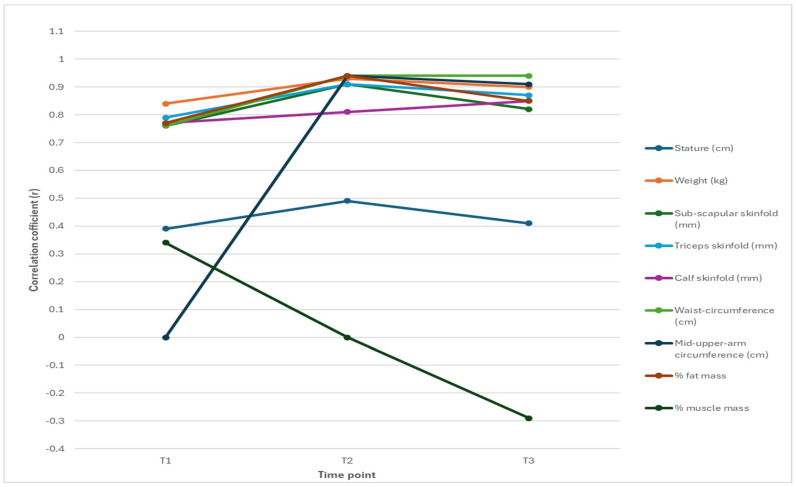
Temporal evolution of BMI correlations in boys.

**Figure 2 jfmk-10-00320-f002:**
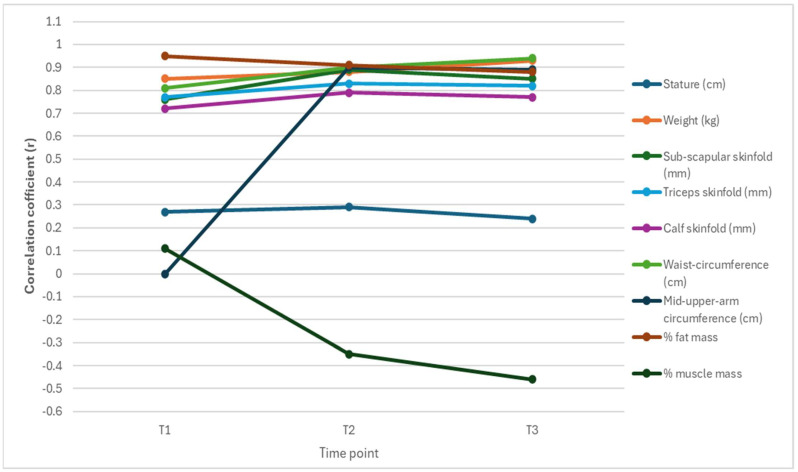
Temporal evolution of BMI correlations in girls.

**Figure 3 jfmk-10-00320-f003:**
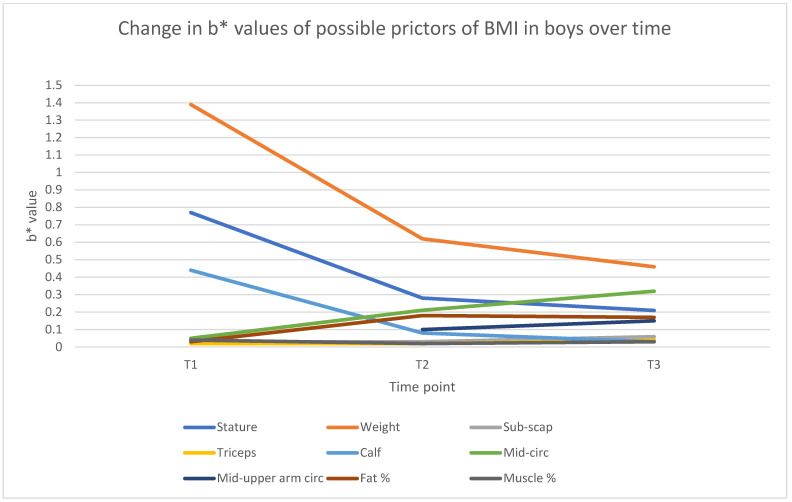
Changes in boys’ predictive b* value over time.

**Figure 4 jfmk-10-00320-f004:**
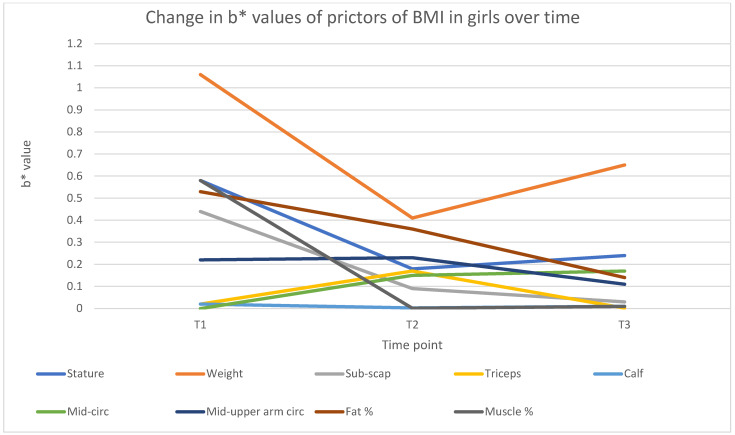
Changes in girls’ predictive b* value over time.

**Table 1 jfmk-10-00320-t001:** Association between BMI and fat percentage scores obtained from the Omrom and mathematical equations.

**BMI (Equations)**			
	**T1**	**T2**	**T3**
BMI(BIA) T1	0.93 ***	-	-
BMI(BIA) T2	-	0.97 ***	-
BMI(BIA) T3	-	-	0.99 ***
**Sum of Skinfolds**			
	**T1**	**T2**	**T3**
Fat % (BIA) T1	0.68 ***	-	-
Fat% % (BIA) T2	-	0.89 ***	-
Fat% % (BIA) T3	-	-	0.86 ***

BMI = body weight index; BIA=Results obtained by bioelectrical impedance analysis; T1 = Baseline measurement (2010); T2 = first follow-up measurement (2013); T3 = second follow-up measurement (2016) 0; *** = r > 0.5 (strong correlation).

**Table 2 jfmk-10-00320-t002:** Basic statistics and significance of changes in the body composition profiles of boys and girls over three time-points.

**Boys (N = 160)**						
**Variables**	**T1**	**T2**	**T3**	**T1-T2**	**T2-T3**	**T1-T3**
Age (years)	6.93 ± 0.51	9.93 ± 0.36	12.93 ± 0.38	3 *	3 *	6 *
BMI	15.95 ± 2.25	17.76 ± 3.49	19.61 ± 4.34	1.81 *	1.85 *	3.66 *
Stature (cm)	120.85 ± 6.49	136.18 ± 6.89	152.60 ± 9.03	15.33 *	16.42 *	31.75 *
Weight (kg)	23.41 ± 5.14	33.25 ± 9.06	46.01 ± 13.87	9.84 *	12.76 *	22.6 *
Sub-scapular sf (mm)	6.29 ± 2.92	7.49 ± 4.92	9.15 ± 7.70	1.2 *	1.66 *	2.86 *
Triceps sf (mm)	8.28 ± 3.64	10.36 ± 5.54	11.05 ± 7.27	2.08 *	0.69	2.77 *
Calf sf (mm)	7.91 ± 3.74	11.81 ± 6.26	13.87 ± 8.53	3.9 *	2.06 *	5.96 *
Waist-circ (cm)	55.13 ± 6.01	61.19 ± 8.16	66.40 ± 10.03	6.06 *	5.21 *	11.27 *
Mid-upper arm circ (cm)	-	20.65 ± 3.70	23.01 ± 4.76	-	2.36 *	-
% Fat Weight	19.09 ± 6.89	20.95 ± 7.84	19.89 ± 8.36	1.86 *	−1.06	0.8
% Muscle Weight	25.23 ± 4.47	31.21 ± 3.03	35.83 ± 4.33	5.98 *	4.62 *	10.6 *
**Girls (N = 172)**						
Age (years)	6.87 ± 0.48	9.87 ± 0.38	12.87 ± 0.37	3 *	3 *	6 *
BMI	15.69 ± 2.02	17.75 ± 3.41	20.28 ± 4.11	2.06 *	2.53 *	4.59 *
Stature (cm)	119.44 ± 5.93	136.35 ± 7.22	154.46 ± 7.08	16.19 *	18.11 *	35.02 *
Weight (kg)	22.51 ± 4.26	33.11 ± 8.16	48.60 ± 11.74	10.6 *	15.49 *	26.09 *
Sub-scapular sf (mm)	7.32 ± 3.14	8.79 ± 5.14	10.73 ± 5.73	1.47 *	1.94 *	3.41 *
Triceps sf (mm)	9.50 ± 3.43	12.48 ± 5.24	14.47 ± 6.92	2.98 *	1.99 *	4.97 *
Calf sf (mm)	9.59 ± 3.65	14.25 ± 5.97	18.67 ± 8.93	4.66 *	4.42 *	9.08 *
Waist-circ (cm)	54.03 ± 4.90	59.51 ± 7.25	66.21 ± 8.85	5.48 *	6.7 *	12.18 *
Mid-upper arm circ (cm)	-	20.81 ± 3.43	23.88 ± 4.70	-	3.07 *	-
% Fat mass	16.41 ± 6.93	21.80 ± 8.23	25.45 ± 7.99	5.39 *	3.65 *	9.04 *
% Muscle mass	26.62 ± 3.05	30.55 ± 2.60	32.71 ± 3.75	3.93 *	2.16 *	6.09 *

BMI = Body mass index; cm-centimeter; kg = kilogram; mm = millimeter; sf = skinfold; circ = circumference; SD = Standard deviation; N = Number of participants; T1 = Baseline measurement, 6 years; (2010); T2 = 9 years (2013); T3= 12 years (2016); * = Statistical significant (*p* < 0.01) Note: Values represent the absolute mean differences.

**Table 3 jfmk-10-00320-t003:** Changes in correlations between various body composition co-variates and BMI and weight of boys and girls over three time-points.

Variable	T1		T2		T3	
	Weight	BMI	Weight	BMI	Weight	BMI
**Boys**						
Stature (cm)	0.78 ***	0.39 **	0.74 ***	0.49 **	0.70 ***	0.41 **
Weight (kg)	-	0.84 ***	-	0.93 ***	-	0.90 ***
Sub-scapular skinfold (mm)	0.70 ***	0.76 ***	0.88 ***	0.91 ***	0.72 ***	0.82 ***
Triceps skinfold (mm)	0.84 ***	0.79 ***	0.89 ***	0.91 ***	0.75 ***	0.87 ***
Calf skinfold (mm)	0.82 ***	0.77 ***	0.78 ***	0.81 ***	0.75 ***	0.85 ***
Waist-circumference (cm)	0.85 **	0.76 ***	0.94 ***	0.94 ***	0.89 ***	0.94 ***
Mid-upper-arm circumference (cm)	-	-	0.94 ***	0.94 ***	0.87 ***	0.91 ***
% fat mass	0.61 ***	0.77 ***	0.83 ***	0.94 **	0.65 ***	0.85 ***
% muscle mass	0.66 ***	0.34 **	0.20 *	0.00	−0.11 *	−0.29 *
**Girls**						
Stature (cm)	0.73 ***	0.27 *	0.62 ***	0.29 **	0.55 ***	0.24 *
Weight (kg)	-	0.85 ***	-	0.88 ***	-	0.93 ***
Sub-scapular skinfold (mm)	0.62 ***	0.76 ***	0.84 ***	0.89 ****	0.78 ***	0.85 ***
Triceps skinfold (mm)	0.75 ***	0.77 ***	0.83 ***	0.83 ***	0.74 ***	0.82 ***
Calf skinfold (mm)	0.70 ***	0.72 ***	0.76 ***	0.79 ***	0.71 ***	0.77 ***
Waist-circumference (cm)	0.81 ***	0.81 ***	0.91 ***	0.90 ***	0.91 ***	0.94 ***
Mid-upper-arm circumference (cm)	-	-	0.91 ***	0.90 ***	0.88 ***	0.89 ***
% fat mass	0.71 ***	0.95 ***	0.82 ***	0.91 ***	0.78 ***	0.88 ***
% muscle mass	0.53 ***	0.11 *	−0.10 *	−0.35 **	−0.33 **	−0.46 **

BMI = Body mass index; sf = skinfold; circ = circumference; T1 = Baseline measurement (2010); T2 = first follow-up measurement (2013); T3 = second follow-up measurement (2016); * = 0.1 < r < 0.3 (small correlation); ** = 0.3 < r < 0.5 (medium correlation); *** = r > 0.5 (strong correlation). Note: Correlations are based on the absolute values of boys and girls.

**Table 4 jfmk-10-00320-t004:** Multiple regression analysis of co-variants in predicting BMI of boys over three time-points.

	**T1 (6 years)** R = 0.945, R^2^ = 0.894 Adjusted R^2^ = 0.889, F(8,151) = 160.32 *p* < 0.000 * Std. error of estimate: 0.748		
Intercept	b = 32.554 Std. error of b = 3.158, *p*-value = 0.000 *		
Variables	b*	Std. error	*p*
Weight (kg)	1.398 *	0.110	<0.01 *
Stature (M)	−0.772 *	0.098	<0.01 *
Waist circumference (cm)	0.052	0.052	0.445
Muscle %	0.044	0.060	0.560
Calf sf (mm)	0.044	0.057	0.726
Fat %	−0.034	0.059	0.322
Sub-scap sf (mm)	−0.030	0.054	0.568
Triceps sf (mm)	−0.024	0.068	0.726
Mid-upper arm circumference (cm)	-	-	-
	**T2 (9 years)** R = 0.987, R^2^ = 0.975 Adjusted R^2^ = 0.974, F (9,150) = 664.95 *p* < 0.000 * Std. error of estimate: 0.562		
	b = 18.969 Std. error of b = 2.003, *p*-value = 0.000 *		
Weight (kg)	0.625 *	0.068	<0.01 *
Stature (cm)	−0.285 *	0.036	<0.01 *
Waist circumference (cm)	0.211 *	0.049	<0.01 *
Fat %	0.185 *	0.039	<0.01 *
Mid-upper arm circumference (cm)	0.105 *	0.053	<0.01 *
Calf sf (mm)	0.084	0.025	<0.01 *
Sub-scap sf (mm)	−0.031	0.039	0.42
Triceps sf (mm)	0.024	0.044	0.57
Muscle %	0.016	0.020	0.41
	**T3 (12 years)** R = 0.981, R^2^ = 0.964 Adjusted R^2^ = 0.961, F (9,150) = 446.74 *p* < 0.000 * Std. error of estimate: 0.848		
	B = 12.520 Std. error of b = 2.105, *p*-value = 0.000 *		
Weight (kg)	0.469 *	0.051	<0.01 *
Waist circumference (cm)	0.323 *	0.046	<0.01 *
Stature (cm)	−0.210 *	0.031	<0.01 *
Fat %	0.171 *	0.042	<0.01 *
Mid-upper arm circumference (cm)	0.154 *	0.040	<0.01 *
Sub-scap sf (mm)	0.066 *	0.030	<0.09 *
Triceps sf (mm)	−0.049	0.047	0.29
Calf sf (mm)	0.039	0.041	0.34
Muscle %	0.031	0.020	0.13

BMI = Body mass index; sf = skinfold; Circ = circumference; T1 = Baseline measurement (2010); T2 = first follow-up measurement (2013); T3 = second follow-up measurement (2016); std. error = standard error; *p* = statistically significant (*p* < 0.05); b* = beta coefficients; * = statistically significant (*p* < 0.05).

**Table 5 jfmk-10-00320-t005:** Multiple regression analysis of co-variants in predicting BMI of girls.

	**T1 (6 years)** R = 0.996 R^2^ = 0.992, Adjusted R^2^ = 0.991, F(8,163) = 263.45 *p* < 0.0000 * Std. error of estimate: 0.18146		
Intercept	b = 25.995 Std. error of b = 0.737, *p*-value = 0.000 *		
Variables	b*	Std. error	*p*
Weight (kg)	1.061 *	0.030	<0.01 *
Stature (cm)	−0.585 *	0.024	<0.01 *
Fat %	0.219 *	0.020	<0.01 *
Muscle %	0.053 *	0.014	<0.01
Sub-scapular (mm)	0.044 *	0.014	<0.01 *
Calf sf (mm)	0.022	0.014	0.11
Triceps sf (mm)	−0.020	0.015	0.17
Waist circ (cm)	−0.000	0.015	0.97
Mid-upper arm circ (cm)	-	-	-
	**T2 (9 years)** R = 0.959 R^2^ = 0.919 Adjusted R^2^ = 0.915, F (9,162) = 206.81 *p* < 0.000 * Std. error of estimate: 0.993		
	B = 12.070 Std. error of b = 2.467, *p*-value = 0.000 *		
Weight (kg)	0.412 *	0.084	<0.01 *
Fat %	0.355 *	0.068	<0.01 *
Mid-upper arm circ (cm)	0.231 *	0.072	<0.01 *
Stature (cm)	−0.178 *	0.041	<0.01 *
Triceps sf (mm)	−0.173 *	0.061	<0.01 *
Waist circ (cm)	0.151 *	0.070	<0.01 *
Sub-scapular sf (mm)	0.092	0.062	0.14
Calf sf (mm)	0.003	0.045	0.94
Muscle %	0.000	0.036	0.98
	**T3 (12 years)** R = 0.988 R^2^ = 0.976 Adjusted R^2^ = 0.975, F (9,162) = 744.33 *p* < 0.000 * Std. error of estimate: 0.649		
	B = 20.688 Std. error of b = 2.049, *p*-value = 0.000 *		
Weight (kg)	0.656 *	0.054	<0.01 *
Stature (cm)	−0.243 *	0.021	<0.01 *
Waist circ (cm)	0.171 *	0.036	<0.01 *
Fat %	0.139 *	0.033	<0.01 *
Mid-upper arm circ (cm)	0.114 *	0.030	<0.01 *
Sub-scapular sf (mm)	0.037	0.026	0.16
Calf sf (mm)	0.014	0.027	0.59
Muscle %	0.009	0.015	0.51
Triceps sf (mm)	0.002	0.032	0.93

BMI = Body mass index; sf = skinfold; Circ = circumference; T1 = Baseline measurement (2010); T2 = first follow-up measurement (2013); T3 = second follow-up measurement (2016); std. error = standard error; *p* = statistically significant (*p* < 0.05); b* = beta coefficients; * = statistically significant (*p* < 0.05).

## Data Availability

The authors confirm that the data supporting these findings are not available online but are available from the authors upon reasonable request, in accordance with NWU policy guidelines.

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
