# Peer review of "Age-Related Changes in Predictors of BMI in 6, 9 and 12-Year-Old Boys and Girls: The NW-CHILD Study"

_jfmk, 2025, doi:10.3390/jfmk10030320_

Round 1
Reviewer 1 Report
Comments and Suggestions for Authors
- The study is interesting, but it lacks statistical rigor in the comparisons presented in Table 2. While it does highlight which covariates are most relevant to BMI—particularly mass—it falls short in tying these findings back to the central issue raised in the introduction: the limitations in assessing fluctuations in BMI over time. Strengthening this connection would enhance the coherence and impact of the study.
- The abstract could provide a clearer explanation of the concept of BMI covariates.
- In line 19: the phrase "BMI from 6 to 12 years" seems inconsistent, as the upper age limit of the study was 13 years. It would be more accurate to state "from 6 to 13 years" if participants up to 13 were included.
- In line 69: the term "BMIz" is introduced, but the meaning of the "z" is not explained.
- Although the introduction highlights the article’s concern with understanding how BMI fluctuates over time, it is not sufficiently didactic. The limitations of the BMI metric in assessing changes over time are only mentioned in general terms. It would strengthen the introduction to include specific examples or numerical illustrations of these limitations.
- The discussion on covariates of BMI across different ages and between boys and girls is clearly presented. However, the final part of the introduction introduces a new point — “The accuracy of a BIA method to determine BMI and Fat% in 93 children was also analyzed...” — without a clear connection to the previous content. This sub-objective appears abruptly and is not sufficiently explained or integrated into the overall rationale of the study.
- It should be clarified whether the selection of districts and schools was randomized to ensure proper sampling. If this was the case, it should be explicitly stated in the methods section. It is important to remember that the sample should be representative of the target population. I am asking this because you mention a random and stratified sampling method to select participants, but it is unclear whether the same method was applied to select the districts and schools.
- Create a timeline figure showing the recruitment, selection, and dropouts of the children.
- You already have data from the same children at three time points of the study. Why didn’t you take advantage of Table 2 (descriptive) to perform comparisons using a repeated measures ANOVA? You’re missing the opportunity to use a table that already has a layout suitable for comparisons over time.
- The section presenting the data from Table 1 is very poorly explained. The variables are confusing, and there is a statistical error in stating “we wanted to determine the agreement” (line 193), when in fact what was performed was a correlation analysis. These are fundamentally different concepts. If the aim was truly to assess agreement, a comparison test or a Bland-Altman analysis should have been conducted instead.
- From lines 213 to 238, the text becomes almost childish, as the authors discuss differences without presenting any p-values or statistical results. It is not acceptable to claim that differences exist without supporting the statement with corresponding p-values or statistical evidence.
- Given that the article is concerned with temporal evolution, wouldn't it be relevant to use the BMI association values from Table 3 to create a figure—something similar to what was done in Figures 1 and 2? Such a figure could uncover important patterns, showing whether BMI gains or loses predictive power at specific developmental stages, thereby providing a more nuanced and informative understanding of its role over time.
- I suggest the authors provide an explanation of how to interpret the β (beta) coefficient derived from the multiple regression analysis. This value plays an important role in the study, and its proper interpretation is essential for understanding the magnitude and direction of the associations being reported.
- Why does the legend in Figures 1 and 2 include "Mass 2"? Was this a mistake?
Reviewer 2 Report
Comments and Suggestions for Authors
The study addresses a relevant topic and presents valuable data; however, the manuscript would benefit from improvements in organization, clarity, and conciseness, particularly in the Results section. I encourage the authors to revise the structure and formatting to enhance readability and align with reporting standards.
Below, I provide a series of comments that may help improve the paper:
-
Page 2, line 61: I suggest that the authors update the manuscript using people-first language throughout. For example, replace "obese children" with "children with obesity".
-
Page 3, line 95: Did the authors receive any funding from the industry mentioned? What was the rationale for selecting this specific equipment?
-
The Introduction is very well written and provides a solid background for understanding the research problem.
Methods
-
I suggest that item 2.2 of the Methods section be relocated to the Results section. Major reporting guidelines recommend that such information be presented in the results.
-
The authors need to clearly state the inclusion, non-inclusion, and exclusion criteria used in the study.
-
Page 4, line 142: How was the fat percentage determined?
-
Which formula was used to estimate body fat percentage?
Results
-
The authors should begin this section by describing the composition of the sample. They mention that individuals without complete data for all time points were excluded—thus, I recommend including a flowchart.
-
Table titles should be written in uppercase letters.
-
The Results section is currently disorganized. I recommend reorganizing the layout of the text and tables for better structure.
-
In several instances, the authors report values with up to eight decimal places. This appears unnecessary and does not contribute to the clarity or precision of the report. I suggest rounding values to two or three decimal places.
-
Overall, the Results section is excessively long. This impairs readability and clarity. I recommend condensing the section to enhance cohesion.
Discussion
Although somewhat lengthy, the Discussion is adequate and seems to contribute meaningfully to the interpretation of the findings.
Minor point
-
Page numbering is incorrect, particularly after the Results section.
Reviewer 3 Report
Comments and Suggestions for Authors
The manuscript presented is interesting and has some strengths, such as the analysis of BMI predictors in children using a longitudinal approach, as well as performing an analysis according to gender. Its results are interesting, but it still has some weaknesses, which I will outline below.
1. Regarding the introduction, I believe that the objective of the study should be emphasized more and less focus should be placed on the measurement methods and their limitations. I also think it would be appropriate to address the effect of the covariates studied in greater detail.
2. Regarding the study design, other variables involved in BMI were not taken into account, such as caloric intake, physical activity, and pubertal status in girls.
3. Regarding the statistical analysis, further explanation is needed, for example, regarding residuals and heteroscedasticity. The authors also do not specify why they did not perform data imputation to avoid excluding missing data.
4. I suggest that the authors discuss more extensively the effect of the increase in fat percentage in girls due to the effect of puberty or other factors.
5. Evaluate whether it is necessary to include reference 23, as it is a preprint.
Reviewer 4 Report
Comments and Suggestions for Authors
This study focuses on body composition, which is important for understanding children's growth, development, and future health outcomes. Specifically, it aims to analyze, from a longitudinal perspective between ages 6 and 12, the factors (covariates) that influence fluctuations in BMI (Body Mass Index). Although BMI is widely used, it has limitations when applied to growing children, so clarifying the factors that influence its changes is considered helpful for more appropriate health assessments and interventions. There is no disagreement on this point.
On the other hand, it is recognized that some aspects require further discussion.
- While recognizing the limitations of BMI, it feels like the focus remains centered on BMI. Although the "limitations of BMI" are acknowledged in the background, BMI is ultimately set as the primary outcome in the conclusion. Are there no more effective alternative indicators? There are expectations for future practical applications in society.
- Despite being a longitudinal study, there is no mention of attrition rates or data completeness. If the number of participants varies at each measurement point, could this introduce bias? I would like to see a discussion regarding this issue.
- It seems that important factors affecting children's BMI and body fat percentage (such as eating habits, physical activity levels, and economic status) have not been taken into consideration. I feel that this should be addressed in future discussions—what do you think?
- This is an analysis set in the unique cultural and nutritional environment of South Africa, but there may be region-specific social factors at play (e.g., urban/rural disparities, nutritional status). Measures for generalization have not been considered. What kind of recommendations are possible?
- The results for girls are said to be "related to fat from an early stage," but it seems that there isn’t sufficient mention of the influence of pubertal sex hormones or differences in the onset of secondary sexual characteristics. What are your thoughts on that point?
Reviewer 5 Report
Comments and Suggestions for Authors
This manuscript reports a correlational analysis to determine sex differences in covariates of BMI in a longitudinal study of 332 children from 6-12 years.
The manuscript needs attention to the English.
To address concerns of representation bias, the authors need to report the baseline characteristics of the children who consented but did not provide data at the three time points versus the 332 who did.
The authors rely in part on Bioimpedance analysis. I am aware of numerous concerns about the validity of this procedure. See below:
Camila E Orsso 1, Maria Cristina Gonzalez 2 3, Michael Johannes Maisch 4, Andrea M Haqq 5, Carla M Prado Using bioelectrical impedance analysis in children and adolescents: Pressing issues Eur J Clin Nutr. 2022 May;76(5):659-665. doi: 10.1038/s41430-021-01018-w.Epub 2021 Oct 7. PMID: 3462099
H Talma 1, M J M Chinapaw, B Bakker, R A HiraSing, C B Terwee, T M Altenburg Bioelectrical impedance analysis to estimate body composition in children and adolescents: a systematic review and evidence appraisal of validity, responsiveness, reliability and measurement error Obes Rev. 2013 Nov;14(11):895-905. doi: 10.1111/obr.12061. Epub 2013 Jul 12.
The authors need to report how they estimated percentage body fat from the BIA readings. These validity concerns should be addressed in a Limitations section.
Table 1: BMI is not a criterion measure, so the correlations in the top part of the table do not establish (lines 204-5): “Therefore, the body composition data obtained from the Omron BF511 can be considered accurate and reliable.” They appear to establish that whatever the BMI measures so does the Omron. Since there have been concerns about BMI, perhaps these concerns extend to the Omron?
Table 2: is “body mass” the child’s weight? Since mass= density x volume = weight/gravity, wouldn’t it be more precise to just call this weight?
The correlations between % fat mass and BMI do not seem to be the same/similar in tables 1 and 3?
The authors report they are analyzing changes in BMI. If they were analyzing changes, their regression equations would need to include the BMI at the previous assessment time point.
The authors report: “Our findings confirmed body mass and body stature as the strongest determinants of BMI between the ages of 6 and 12 years, and especially at 6 years of age.” At other points they report influences on BMI. These are terms of causality; the authors, however, only have correlations. BMI is considered a rough index of body composition which has been very controversial over the years, and even more so recently. As a rough index, it might be appropriate to talk about what BMI reflects at each time, but these analyses do not arise to the level of causality.
All these variables are highly intercorrelated, likely arising to a serious concern for multicollinearity: “multicollinearity is a situation where variables are so closely related that it becomes difficult to isolate their individual effects on the dependent variable.” The authors may wish to conduct a principal components analysis of the variables, other than mass and stature, to determine if they can find somewhat less related factors?
The fact that mass and stature correlate highly with BMI is simply a fact that BMI is calculated from mass and stature. It’s the same thing. You are correlating the index with components of itself. In fact it is disappointing that the correlations were so low.
Comments on the Quality of English LanguageThere are several places where there are disfluencies in the English.
Round 2
Reviewer 1 Report
Comments and Suggestions for Authors
The article has improved a lot, but I still have a few small suggestions. Please clarify the legend of Table 3—make it clearer that the absolute differences between the teams are being reported. Figures 3 and 4 look very compressed; the axes and overall graphic quality need to be improved. The conclusion is quite long compared to standard formats—consider shortening it. Also, in the discussion, try to include some kind of summary of the main findings.
Reviewer 3 Report
Comments and Suggestions for Authors
I agree with the modifications made by the authors.
Reviewer 4 Report
Comments and Suggestions for Authors
The authors have duly revised and expanded on my previous comments. I have no further revisions to make.
